# Neuroprotective Actions of Hydrogen Sulfide-Releasing Compounds in Isolated Bovine Retinae

**DOI:** 10.3390/ph17101311

**Published:** 2024-10-01

**Authors:** Leah Bush, Jenaye Robinson, Anthonia Okolie, Fatima Muili, Catherine A. Opere, Matthew Whiteman, Sunny E. Ohia, Ya Fatou Njie Mbye

**Affiliations:** 1Department of Pharmaceutical Sciences, College of Pharmacy and Health Sciences, Texas Southern University, Houston, TX 77004, USA; lemwjb@gmail.com (L.B.); jenayerobinson@yahoo.com (J.R.); a.okolie9552@student.tsu.edu (A.O.); f.muili1867@student.tsu.edu (F.M.); sunny.ohia@tsu.edu (S.E.O.); 2Department of Pharmacy Sciences, School of Pharmacy and Health Professions, Creighton University, Omaha, NE 68178, USA; catherineopere@creighton.edu; 3University of Exeter Medical School, St. Luke’s Campus, Magdalen Road, Exeter EX1 2LU, UK; m.whiteman@exeter.ac.uk

**Keywords:** hydrogen sulfide, retina, oxidative stress, lipid peroxidation

## Abstract

**Background:** We have evidence that hydrogen sulfide (H_2_S)-releasing compounds can reduce intraocular pressure in normotensive and glaucomatous rabbits by increasing the aqueous humor (AH) outflow through the trabecular meshwork. Since H_2_S has been reported to possess neuroprotective actions, the prevention of retinal ganglion cell loss is an important strategy in the pharmacotherapy of glaucoma. Consequently, the present study aimed to investigate the neuroprotective actions of H_2_S-releasing compounds against hydrogen peroxide (H_2_O_2_)-induced oxidative stress in an isolated bovine retina. **Materials and Methods:** The isolated neural retinae were pretreated with a substrate for H_2_S biosynthesis called L-cysteine, with the fast H_2_S-releasing compound sodium hydrosulfide, and with a mitochondrial-targeting H_2_S-releasing compound, AP123, for thirty minutes before a 30-min oxidative insult with H_2_O_2_ (100 µM). Lipid peroxidation was assessed via an enzyme immunoassay by measuring the stable oxidative stress marker, 8-epi PGF2α (8-isoprostane), levels in the retinal tissues. To determine the role of endogenous H_2_S, studies were performed using the following biosynthesis enzyme inhibitors: aminooxyacetic acid (AOAA, 30 µM); a cystathione-β-synthase/cystathionine-γ-lyase (CBS/CSE) inhibitor, α–ketobutyric acid (KBA, 1 mM); and a 3-mercaptopyruvate-s-sulfurtransferase (3-MST) inhibitor, in the absence and presence of H_2_S-releasing compounds. **Results:** Exposure of the isolated retinas to H_2_O_2_ produced a time-dependent (10–40 min) and concentration-dependent (30–300 µM) increase in the 8-isoprostane levels when compared to the untreated tissues. L-cysteine (10 nM–1 µM) and NaHS (30 –100 µM) significantly (*p* < 0.001; n = 12) prevented H_2_O_2_-induced oxidative damage in a concentration-dependent manner. Furthermore, AP123 (100 nM–1 µM) attenuated oxidative H_2_O_2_ damage resulted in an approximated 60% reduction in 8-isoprostane levels compared to the tissues treated with H_2_O_2_ alone. While AOAA (30 µM) and KBA (1 mM) did not affect the L-cysteine evoked attenuation of H_2_O_2_-induced oxidative stress, KBA reversed the antioxidant responses caused by AP123. **Conclusions:** In conclusion, various forms of H_2_S-releasing compounds and the substrate, L-cysteine, can prevent H_2_O_2_-induced lipid peroxidation in an isolated bovine retina.

## 1. Introduction

Lipid peroxidation-induced damage has been implicated in the pathogenesis of a variety of neurodegenerative diseases [1,2]. Evidence from the literature indicates that oxidative stress (OS) also plays a major role in ocular pathologies such as cataracts, age-related macular degeneration (ARMD), glaucoma, and diabetic retinopathy (DR) [3,4,5,6]. Under normal physiological conditions, the presence of several intrinsic antioxidant enzymes in the eye prevents OS formation as a consequence of normal cellular metabolism [2,7,8]. Hydrogen peroxide (H_2_O_2_) is a biologically derived oxidant intermediate which acts with other peroxide-induced free radicals to inflict damage on ocular tissues by changing cellular structure and function [9]. Because of its high content of polyunsaturated fatty acids, the retina is highly susceptible to damage via free radical oxidation [2,10]. Products of the oxygen-derived free radical pathways, including peroxides and long-chain polyunsaturated fatty acid (LCPUFA) metabolites such as isoprostanes, contribute to oxidative reactions in the eye and are important in the pathophysiology of most ocular diseases [2,11,12,13,14]. Indeed, isoprostanes are involved in the pharmacological actions of H_2_O_2_ in ocular tissues and have been shown to regulate sympathetic and excitatory amino acid neurotransmission in the eye [2,15]. The use of pharmacological agents to prevent OS could be an effective strategy in protecting ocular tissues such as the retina from oxidative damage.

Hydrogen sulfide (H_2_S) is a colorless gas that has long been recognized as an environmental pollutant and a highly toxic gas [16]. Since the discovery of its basal production in mammalian tissues about two decades ago, H_2_S has assumed the role of a gaseous mediator with biological importance in the central and peripheral nervous systems [17,18,19,20,21]. This novel gaseous transmitter, H_2_S, is endogenously produced in mammalian tissues from the amino acids L-cysteine and homocysteine in the presence of two pyridoxal-5-phosphate-dependant enzymes, cystathionine b-synthase (CBS), and cystathionine γ-lyase (CSE). 3-mercaptopyruvate sulfurtransferase (3-MST), has been reported to be involved in the calcium-dependent production of H_2_S [22,23,24]. In recent years, d-amino oxidase (DAO) has been identified as the fourth enzyme involved in the biosynthesis of H_2_S, as it is responsible for the conversion of d-cysteine into H_2_S through the 3-MST/CAT pathway [25,26]. As research into the biological activities of H_2_S evolved, the sources of H_2_S expanded from the creation of fast- and slow-releasing H_2_S compounds such as sodium hydrosulfide (NaHS) and GYY4137, respectively, to the development of H_2_S donors targeted at the mitochondrial sources of the gas such as AP123 and AP39 [27,28,29]. Indeed, H_2_S has been reported to be involved in several physiological and pathophysiological processes such as learning and memory, cell survival, inflammation, modulation of synaptic activities in the central nervous system, and maintenance of vascular tone in the cardiovascular system [20,30,31,32,33,34]. Because of the localization of the enzymes responsible for the biosynthesis of H_2_S in cells and tissues, the endogenously produced gas has been shown to mediate biological processes [22,35,36]. In the eye, H_2_S is present in the retina [37] and has been shown to exert pharmacological actions such as the inhibition of excitatory amino acid neurotransmission [38] and increases in cyclic AMP formation [39,40]. The presence of H_2_S in the retina and its transsulfuration pathways indicates a role for this gas in several physiological processes that affect cellular signaling and redox homeostasis [41].

As a retinal neurodegenerative disease, current strategies for the treatment of glaucoma include using agents that can provide neuroprotection for the ganglion cell layer [42]. The gaseous neurotransmitter, H_2_S, has been reported to play a role in several neurodegenerative diseases such as Parkinson’s disease, Alzheimer’s disease, Amyotrophic Lateral Sclerosis, and Down Syndrome [43]. There is evidence that H_2_S can prevent ischemia/reperfusion injury and promote retinal glial cell survival indicating a possible role for this gas in glaucoma-related damage [44]. While there is evidence that H_2_S is produced in the eye and can protect the retina from light- and NMDA-induced neurodegenerative damage [22,45], the effect of this gas on LCPUFA metabolites and lipid peroxidation in the retina is unknown. The aim of the present study was, therefore, to investigate the pharmacological actions of H_2_S derived from several sources (a substrate for enzymatic H_2_S biosynthesis, L-cysteine; a fast-releasing H_2_S compound, NaHS; and a mitochondrial-targeting H_2_S compound, AP123) against oxidative damage induced by H_2_O_2_ in an isolated bovine neural retina.

## 2. Results

### 2.1. Effect of H_2_O_2_ on 8-Isoprostane Production in the Retina

Isoprostanes are stable compounds whose production increases with exposure to OS, thus making them dependable markers of oxidative damage in both in vivo and in vitro animal models [46,47]. In a series of experiments, we studied the effects of time and different concentrations of H_2_O_2_ on the production of 8-isoprostane in an isolated bovine neural retina. As illustrated in Figure 1, 10–40 min insults with H_2_O_2_ (30–300 µM) elicited a concentration- and time-dependent increase in the 8-isoprostane levels (Figure 1). There was a time-dependent increase in the basal 8-isoprostane concentrations, reaching a maximum at 40 min. The effects elicited by H_2_O_2_ (100 µM) also showed a time-dependent increase reaching a maximum at 40 min. As a result of the data obtained in the studies described in Figure 1, all subsequent experiments assessing the pharmacological actions of H_2_S-producing compounds against H_2_O_2_-induced damage on isolated bovine retinae were performed in the presence of H_2_O_2_ (100 µM) for 30 min.

### 2.2. Effects of H_2_S-Releasing Compounds on Lipid Peroxidation in the Bovine Retina

Although evidence in the literature supports a protective role for H_2_S against OS in neurons in vivo [46,48], the neuroprotective action of this gas in ocular tissues under conditions of OS induced by H_2_O_2_ in vitro warrants investigation. To assess the pharmacological role of endogenously produced H_2_S in the presence of the substrate, L-cysteine, isolated retinal tissues were pretreated with L-cysteine (10 nM–1 µM) for 30 min before an insult with H_2_O_2_ (100 µM). L-cysteine (10 nM) significantly (*p* < 0.001; n = 12) reversed the H_2_O_2_-induced increase in the 8-isoprostane levels, producing a 43% reduction in 8-isoprostane generation compared to the control tissues in the presence of H_2_O_2_. (Figure 2). In contrast, higher concentrations of L-cysteine (100 nM–1 µM) do not affect H_2_O_2_-induced damage.

We then examined the effects of a fast-releasing H_2_S compound, NaHS, on 8-isoprostane production in the retina under conditions of H_2_O_2_-induced OS. Treatment of the tissues with NaHS (10–30 µM) for 10 min before the administration of H_2_O_2_ (100 µM) for 30 min significantly (*p* < 0.001; n = 12) attenuated 8-isoprostane production, whereas a high concentration of NaHS (100 µM) significantly (*p* < 0.001; n = 12) augmented the retinal 8-isoprostane levels, approximately 12% over the control (H_2_O_2_ treated tissues) (Figure 3).

To examine the pharmacological action of the novel mitochondrial-targeting H_2_S-releasing compound, AP123, retinal tissues were exposed to varying concentrations of AP123 (10 nM–1 µM) before exposure to H_2_O_2_ (100 µM). After 30 min of treatment, AP123 (10 nM–1 µM) attenuated the H_2_O_2_-induced increase in 8-isoprostane production and even reduced the 8-isoprostane levels in the bovine retinae by 60% at higher concentrations (100 nM–1 µM) compared to the control tissues in the presence of H_2_O_2_. A maximal inhibitory effect on isoprostane production was observed with 100 nM of AP123 (Figure 4).

### 2.3. Effect of Inhibitors of CBS/CSE and 3-MST on the Antioxidant Actions of H_2_S-Producing Compounds

Endogenously produced H_2_S has been shown to protect retinal photoreceptors against light-induced oxidative damage [22]. In the present study, we investigated the role of endogenously produced H_2_S in the neuroprotective action of L-cysteine against H_2_O_2_-induced lipid peroxidation in isolated bovine neural retina. In a series of experiments, we tested the effect of an inhibitor of both CBS and CSE, amino-oxyacetic acid [AOAA, (30 µM)], and an inhibitor of 3-MST, ketobutyric acid [49] (KBA, 1 mM), on the pharmacological effect elicited by L-cysteine. Both AOAA (30 µM) and KBA (1 mM) did not affect the basal 8-isoprostane levels and had no significant effect on the reversal of H_2_O_2_-induced 8-isoprostane production caused by L-cysteine (100 µM) (Figure 5). The evidence in the literature suggests that AP123, the mitochondrial H_2_S donor, can exert antioxidant effects in the microvascular endothelial cells [27]. In the present study, we evaluated the role of endogenous H_2_S production through the 3-MST/H_2_S pathway in the antioxidant activity of AP123 in an isolated bovine retina under the conditions of H_2_O_2_-induced OS. On its own, KBA (1 mM) did not affect basal 8-isoprostane production. In the presence of KBA (1 mM), the inhibitory actions of AP123 on 8-isoprostane generation were significantly (*p* < 0.001; n = 12) reversed (Figure 6).

## 3. Discussion

ROS are formed during normal cellular metabolism, and there is evidence that they play an important, yet complex role in biological systems [1,2]. In excess, ROS can react with DNA, proteins, and lipids to form pharmacologically active metabolites that can perpetuate pathophysiological conditions [1,2]. Isoprostanes are chemically stable, prostaglandin-like lipid peroxidation products that are endogenously formed from oxidative damage to LCPUFAs [50]. These stable molecules are utilized as markers of lipid peroxidation in mammalian tissues, as their levels increase with oxidative damage [51,52]. While the evidence available supports the use of several markers to validate oxidative stress in biological samples [53,54], we selected 8-isoprostanes because of their direct relevance to the measurement of oxidative stress in retinal tissues [55,56,57]. In the present study, we found that various concentrations of H_2_O_2_ elicited a time-dependent increase in 8-isoprostane production in isolated bovine retinal tissues. The ability of exogenously administered H_2_O_2_ to cause a measurable increase in 8-isoprostane production supports the fact that lipid peroxidation occurs in response to this peroxide. The exposure of the tissues to increasing concentrations of H_2_O_2_ (30–300 µM) caused a time- and concentration-related increase in H_2_O_2_-induced lipid peroxidation, reaching a maximum at 300 µM. For this reason, an insult of 30 min duration with a submaximal concentration of H_2_O_2_ (100 µM) was selected as oxidative stimuli for all the experiments involving OS. It is pertinent to note that the concentration of H_2_O_2_ employed for inducing oxidative stress in retinal cells in the present study (100 µM) is much lower than that reported by other investigators such as Cui et al. [58] (200–800 µM), Hu et al. [59] (300–400 µM), and Wang et al. [60] (300 µM).

The biological actions of H_2_S have been documented in various mammalian tissues and systems [61]. This gaseous neurotransmitter is involved in a variety of pathophysiological processes such as inflammation, memory and learning, and in the regulation of blood pressure [16,62]. In the cardiovascular system, H_2_S has been shown to play a vital role in the maintenance of vascular smooth muscle tone, and in the central nervous system, this gas has been found to act as a neurotransmitter at synapses [63,64,65]. Cytoprotection is one of the many roles of this gas, as H_2_S has been reported to exert a neuroprotective action on neurons [48]. In 2011, Biermann and co-workers demonstrated that the exposure of animals to inhalational H_2_S decreased the apoptosis of retinal ganglion cells in a rat model of ischemia/reperfusion injury, suggesting a role for this gas as a neuroprotectant [66]. Because of the inherent challenges involved in the use of H_2_S gas in biological studies, the kinetic profiles of the release of this gas from some organic and inorganic compounds were extensively studied [28,67]. For instance, NaHS was reported to release H_2_S gas rapidly in biological media, whereas GYY4137 is a slow-releasing H_2_S compound [28]. There is evidence that H_2_S-releasing compounds such as NaHS can inhibit both light-induced degeneration and NMDA-induced oxidative damage in the retina, supporting a neuroprotective role for H_2_S [22,41]. H_2_S-releasing compounds have also been reported to inhibit excitatory amino acid neurotransmission (a marker of excitotoxicity) in an isolated bovine retina [38,68,69] and in H_2_O_2_-induced OS and cataract formation in cultured bovine lenses [70].

In the present study, the amino acid substrate for H_2_S production, L-cysteine, the fast-releasing H_2_S compound, NaHS, and the mitochondrial-targeted H_2_S compound, AP123, all protected the neural retina against H_2_O_2_-induced lipid peroxidation. Pretreatment with low concentrations of L-cysteine for 30 min attenuated lipid peroxidation whereas higher concentrations of this amino acid had no such action. The amino acid L-cysteine acts as a substrate for the enzymatic production of H_2_S and has been found to play a developmental and cytoprotective role in the CNS [71,72]. L-cysteine has been reported to promote the proliferation and differentiation of neural stem cells via the H_2_S/CBS pathway [72] and can protect against cellular damage in brain tissue [71]. Indeed, L-cysteine was found to upregulate the expression and activity of CBS and reduce brain edema in rats following a subarachnoid hemorrhage [71]. Taken together, our data suggest that the endogenous production of H_2_S from the substrate L-cysteine can protect the neural retina against H_2_O_2_-induced oxidative damage.

As observed with L-cysteine, pretreatment of tissues with low concentrations of NaHS for 30 min prevented H_2_O_2_-induced lipid peroxidation whereas a higher concentration of this fast-releasing H_2_S compound had no effect on oxidative damage. Interestingly, the pretreatment of tissues with a higher concentration of NaHS significantly increased (*p* < 0.001; n = 12) the levels of this lipid metabolite by 44% when compared with the control tissues. Both NaHS and N-acetyl-cysteine (NAC), a derivative of L-cysteine, have both been shown to exert concentration-related pharmacological actions that can be either protective or deleterious in biological systems [62,73,74]. Depending on the timing and duration of the H_2_S exposure prior to an oxidative insult as well as the concentration of the H_2_S-releasing compound utilized, the pharmacological actions of this gas can vary from cytoprotective and anti-inflammatory [62,71,74] to apoptotic and pro-inflammatory [62,73,74]. NaHS, an inorganic sulfide salt that instantaneously releases H_2_S in an aqueous solution, has been found to protect tissues from oxidative damage [45,66,75]. In a mouse model for diabetes, NaHS (100 µmol/kg) attenuated the production of ROS via the nicotine adenine dinucleotide phosphate (NADPH) oxidase and restored endothelial function in the aorta of diabetic mice [45]. The cortical neurons were protected against glutamate-induced excitotoxicity by NaHS (100 µM) via an observed increase in the activity of glutamate cysteine ligase, an enzyme involved in the production of the antioxidant GSH, and the augmentation of the intracellular GSH levels [75]. It is tempting to speculate that the observed protective effects of NaHS in the present study could be attributed to the stimulatory effect of NaHS on glutamate cysteine ligase and GSH production; however, further studies are needed to confirm this theory.

In the present study, the treatment of tissues with L-cysteine (10 nM) elicited a 43% reduction in the 8-isoprostane levels in the presence of H_2_O_2_, making it the most potent compound tested. The high potency of L-cysteine compared to the other H_2_S-releasing compounds has been observed in its pharmacological actions in ocular tissues. In a 2017 study, L-cysteine (100 nM) was found to increase the aqueous humor outflow facility by 150%, whereas a much higher dose of NaHS (10 µM) was required to exert a similar effect [76]. Similarly, Ohia and colleagues observed that L-cysteine elicited the most potent relaxation of carbachol-induced tone in porcine isolated irides when compared to the H_2_S-releasing compounds such as NaHS and Na_2_S [77].

AP123, a novel mitochondrial-targeting H_2_S-releasing compound, has been found to play a cytoprotective role in the cardiovascular system [27]. Indeed, AP123 (10 nM to 3 µM) was reported to stabilize mitochondrial membrane potential and significantly reduce the generation of ROS in hyperglycemic vascular endothelial cells [27]. In the present study, we found that the neuroprotective actions of AP123 in bovine retinae were observed in a similar concentration range as those demonstrated by Gero et al. [27]. Interestingly, AP123 not only prevented the H_2_O_2_-induced lipid peroxidation but it also elicited a further decrease in the basal isoprostane levels at higher concentrations. Through its actions as an electron donor in the electron transport chain, H_2_S has been found to stabilize membrane potential, thereby inhibiting the production of mitochondrial ROS in the vascular endothelium [78]. It is tempting to speculate that the observed marked effects of AP123 on lipid peroxidation in the present study may be related to its actions in H_2_S biosynthesis at the mitochondrial level.

In another series of experiments, we studied the role of endogenously produced H_2_S in the pharmacological action of L-cysteine against lipid peroxidation. The inhibitors of the enzymes responsible for the biosynthesis of H_2_S, (CBS, CSE, and 3-MST) were ineffective at diminishing the inhibitory effect elicited by L-cysteine on isoprostane production. Taken together, our data suggest that the protective actions of this amino acid in an isolated bovine retina are independent of its de novo enzymatic conversion to H_2_S. It is important to note that L-cysteine is also the rate-limiting agent in the production of glutathione (GSH), which possesses a cytoprotective function in various cell types due to its ROS-scavenging capabilities [8,79]. It may well be that the high potency displayed by L-cysteine in our studies could be related to its additional ability to regulate the production of GSH, a potent antioxidant.

3-MST is an enzyme that is constitutively expressed in mitochondria, and the production of H_2_S through this pathway is neuroprotective in the brain and retina [22,24]. In the present study, inhibition of 3-MST with KBA abolished the pharmacological actions of AP123, suggesting that the endogenous biosynthesis of H_2_S via the 3-MST pathway is involved in the pharmacological action of this H_2_S-releasing compound in an isolated bovine retina. Since AP123 is a mitochondrial-targeting H_2_S-releasing compound and 3-MST is localized to the mitochondria, it appears that the direct delivery of H_2_S to the mitochondria may account, at least in part, for the ability of KBA to effectively prevent the H_2_O_2_-induced lipid peroxidation. In summary, the data obtained using AP123 highlight a role for the mitochondria in serving as an important source of H_2_S in the observed protective action exhibited by some H_2_S-releasing compounds.

H_2_S-releasing compounds have been shown to reduce intraocular pressure in normotensive rabbits [80] and to increase the outflow facility in porcine ocular anterior segments, ex vivo [76], suggesting a potential role for this gas in glaucoma pharmacotherapy [81]. The current observation that H_2_S-releasing compounds can prevent oxidative damage in the retina supports a dual role for this gas in altering aqueous humor dynamics and serving as a neuroprotectant for the eye.

## 4. Materials and Methods

### 4.1. Chemicals

The H_2_O_2_, L-cysteine, sodium hydrosulfide (NaHS), amino-oxyacetic acid (AOAA), and α-ketobutyric acid (KBA) were purchased from Sigma Chemical (St. Louis, MO, USA). GYY4137, the 8-Isoprostane enzyme-linked immunoassay kit, and the protein determination assay kit were purchased from Cayman Chemical (Ann Harbor, MI, USA). AP123 was a gift from Dr. Matthew Whiteman’s laboratory. All test agents were freshly prepared immediately before use in the series of experiments. Stock solutions of AOAA were prepared in 70% ethanol. All other stock solutions were prepared in deionized water.

### 4.2. Tissue Preparation

Studies were performed using bovine eyes procured from a local slaughterhouse and transported to the laboratory in an ice bath. A cut was made along the equator of each eye, and the vitreous humor and lens were delicately dissected out. The neural retina was isolated by a gentle dislocation from the posterior segment of the eye and immediately immersed in a warm, oxygenated Krebs buffer solution (pH 7.4). After the isolation of the neural retina, the tissues were cut into four pieces and randomly assigned to treatment groups. The control groups received neither H_2_O_2_ nor H_2_S-releasing compounds/enzyme inhibitors. The time elapsed between the animal sacrifice and tissue preparation in the present study was less than 24 h; a protocol consistent with observations made by Murali et al. [82] who reported that human retinal tissues remained resilient to cell death, with a substantial number of cells remaining viable for up to five days postmortem.

### 4.3. 8-Isoprostane ELISA Assay

The methodology used for the extraction of 8-isoprostane was essentially the same as described by our laboratory and other investigators [83,84,85,86] with some modifications. The isolated bovine retinae were equilibrated in an oxygenated Krebs solution at 37 °C for 20 min. Tissues were then transferred and incubated in a Krebs solution in the presence and absence of H_2_O_2_ for 10 to 40 min. To determine the pharmacological effect of H_2_S-releasing compounds on H_2_O_2_-induced oxidative stress, the retinal tissues were treated with test compounds for 10–30 min before being subjected to a 30 min exposure to H_2_O_2_. To determine the role of enzymes in the biosynthesis of H_2_S, the retinae were treated with the CBS and CSE inhibitor (AOAA (30 µM)) and the noncompetitive 3-MST inhibitor, α-ketobutyric acid, (KBA (1 mM) to determine the involvement of the enzymes responsible for H_2_S biosynthesis in AP123- and L-cysteine-mediated actions. After incubation, tissues were homogenized in a 0.1 M phosphate buffer (pH 7.4) containing 1 mM of EDTA and 0.005% BHT (1 mL/100 mg tissue) and centrifuged at 3000 rpm for ten minutes at 5 °C. The supernatant was collected and purified using potassium hydroxide (15% *w*/*v*). The 8-isoprotane was extracted from purified samples using solid phase extraction cartridges and an ethyl acetate–methanol (99:1) mixture [87]. The 8-isoprotane was concentrated into a pellet by evaporating the ethyl acetate–methanol solution under N_2_ gas. The enzyme immunoassay buffer was used to re-suspend the 8-isoprotane prior to plating. Protein content was determined from the unpurified supernatant using a Cayman protein determination kit.

### 4.4. Data Analysis

The results were expressed as 8-isoprotane concentrations per milligram of soluble protein (pg/mg protein). Except where indicated, the values given are means ± S.E.M. The significance of differences between the values obtained in the control and drug-treated preparations were evaluated using one-way ANOVA followed by Tukey’s post hoc analysis. The time- and concentration-dependent effects were determined using two-way ANOVA. The differences in *p* values < 0.05 were accepted as statistically significant.

## 5. Conclusions

In conclusion, the substrate for the biosynthesis of H_2_S, L-cysteine, and H_2_S-releasing compounds, NaHS and AP123, can prevent H_2_O_2_-induced OS in an isolated bovine neural retina in a concentration-dependent manner. The pharmacological effect of L-cysteine was independent of the de novo intramural biosynthesis of H_2_S in the retina. On the other hand, the pharmacological action of the mitochondrial-targeted H_2_S-releasing compound, AP123, was mediated by the mechanisms that involve the 3-MST pathway for the biosynthesis of this gas in an isolated bovine retina. Taken together, our data support a neuroprotective role for H_2_S in the retina, a gas that has been reported to play a role in the regulation of aqueous humor dynamics in mammalian eyes.

## Figures and Tables

**Figure 1 pharmaceuticals-17-01311-f001:**
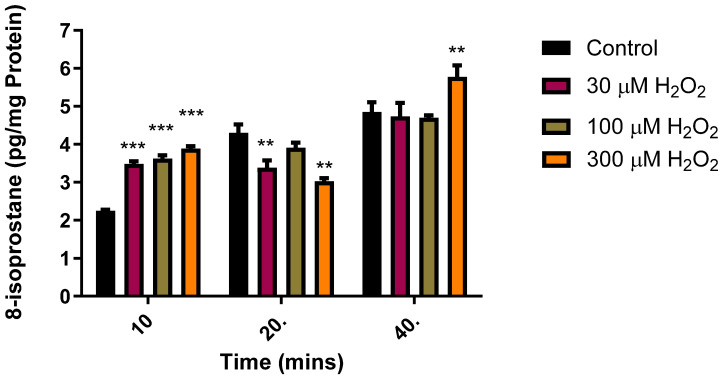
Time- and concentration-dependent effect of H_2_O_2_ on 8-isoprostane production in the bovine retinae. Each value represents the mean ± SEM for n = 12; *** *p* < 0.001 is significantly different from the control and ** *p* < 0.01 is significantly different from the control.

**Figure 2 pharmaceuticals-17-01311-f002:**
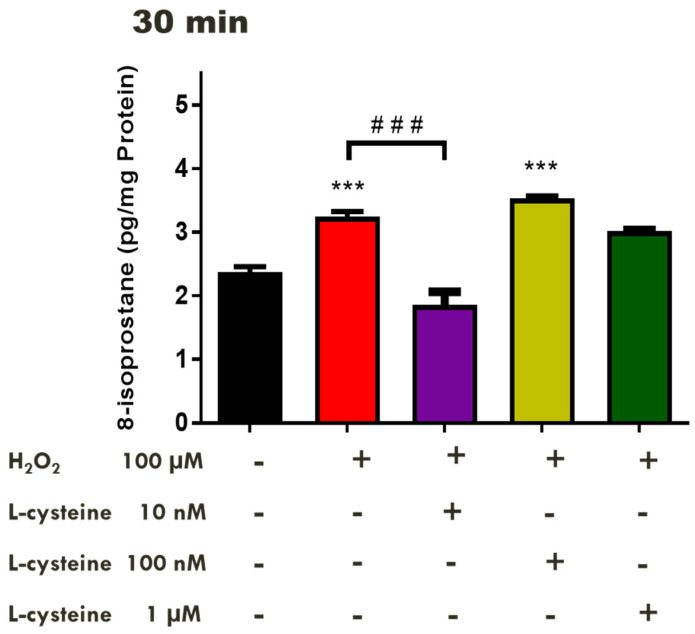
Concentration-dependent effect of L-cysteine on H_2_O_2_-induced oxidative stress in the bovine retinae. Each value represents the mean ± SEM for n = 12; *** *p* < 0.001 is significantly different from the control and ^###^
*p* < 0.001 is significantly different from the H_2_O_2_ treated tissues.

**Figure 3 pharmaceuticals-17-01311-f003:**
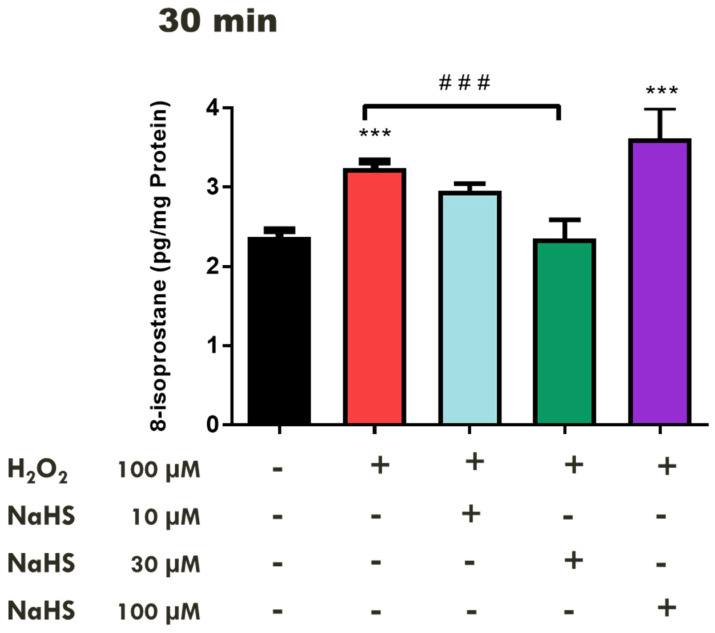
Concentration-dependent Effect of NaHS on H_2_O_2_–induced oxidative stress in the bovine retinae. Each value represents the mean ± SEM for n = 12; *** *p* < 0.001 is significantly different from the control and ^###^
*p* < 0.001 is significantly different from the H_2_O_2_ treated tissues.

**Figure 4 pharmaceuticals-17-01311-f004:**
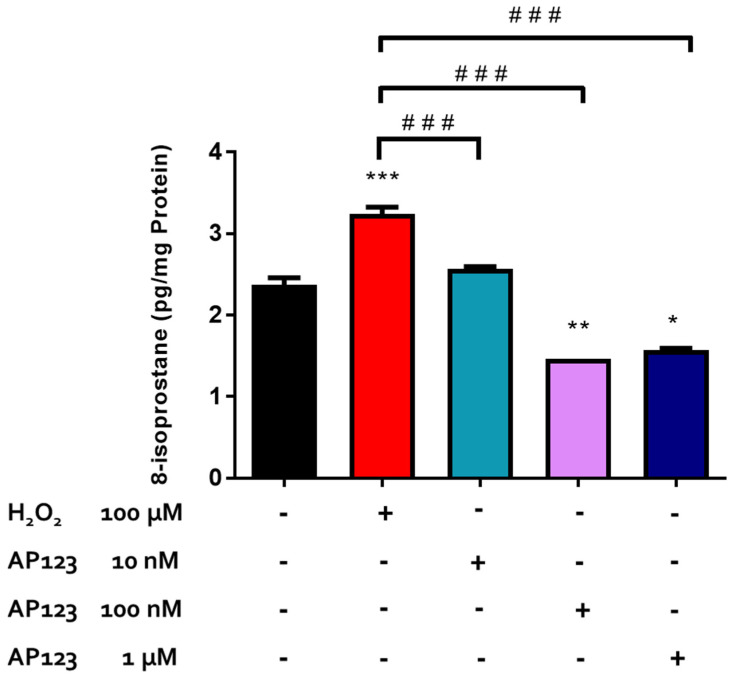
Concentration-dependent effect of AP123 on H_2_O_2_-induced oxidative stress in the bovine retinae. Each value represents the mean ± SEM for n = 12; *** *p* < 0.001 is significantly different from the control, ** *p* < 0.01 is significantly different from the control, * *p* < 0.05 is significantly different from the control, and ^###^ *p* < 0.001 is significantly different from the H_2_O_2_ treated tissues.

**Figure 5 pharmaceuticals-17-01311-f005:**
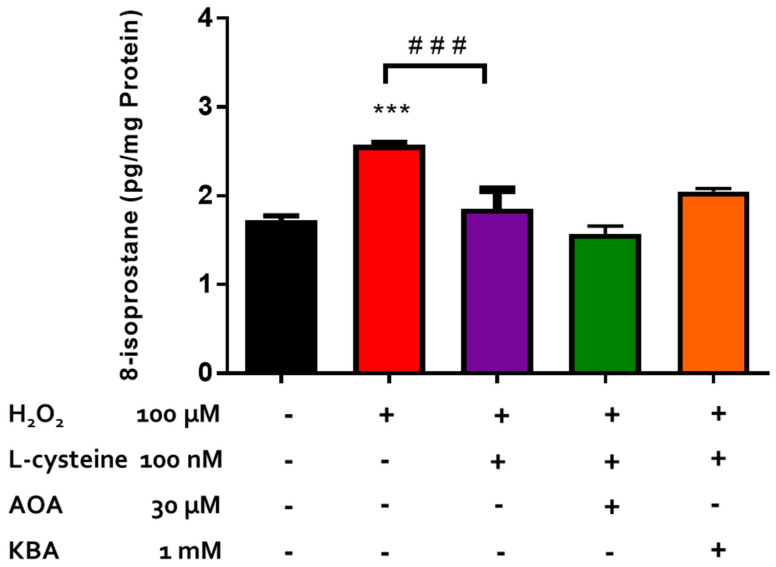
Role of the 3-MST/CAT pathway in AP123-mediated neuroprotection. Each value represents the mean ± SEM for n = 12; *** *p* < 0.001 is significantly different from the control and ^###^ *p* < 0.001 is significantly different from the H_2_O_2_ treated tissues.

**Figure 6 pharmaceuticals-17-01311-f006:**
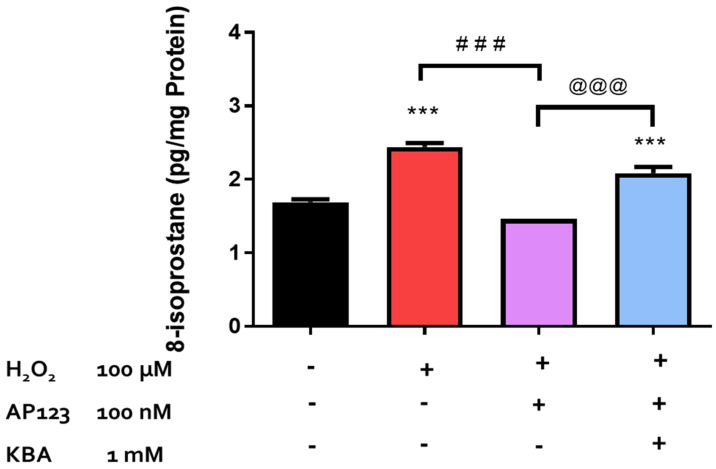
Role of CBS/CSE and the 3-MST/CAT pathway in L-cysteine-mediated neuroprotection. Each value represents the mean ± SEM for n = 12; *** *p* < 0.001 is significantly different from the control, ^###^
*p* < 0.001 is significantly different from the H_2_O_2_ treated tissues, and ^@@@^
*p* < 0.001 is significantly different from the AP123 + H_2_O_2_ treated tissues.

## Data Availability

The data are presented in the main text.

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
