# Peer review of "Neuroprotective Actions of Hydrogen Sulfide-Releasing Compounds in Isolated Bovine Retinae"

_pharmaceuticals, 2024, doi:10.3390/ph17101311_

Round 1

Reviewer 1 Report

Comments and Suggestions for Authors

Leah Bush et al. reported an interesting work upon the effects of H2S on retina. The topic was less frequently discussed yet important. The paper fell within the scope of Pharmaceuticals. The experimental design was logical. However, there were some significant formatting issues pending addressed. Please make a Major Revision before a second decision of this paper.

Please refer to the following comments for formatting issues:

1-      It was confusing to use both upper-case and lower case characters in the Title. Please correct.

2-      The codes of subtitles were incorrect.

3-      The fronts used in figures were not unified.

4-      The size of figures varied a lot.

Please refer to the following scientific comments:

1-      “Percent match: 41%” was revealed by the iThenticate similarity system. Please keep in mind that ethical issues must be avoided. Maybe a total similarity index < 20% should be reached?

2-      Please test the cytotoxicity of the reagents in retina cells.

3-      The H2S production kinetics of the compounds should be evaluated.

4-      What did the “Concentration (p < 0.03) Time (p < 0.0268) Interaction (p < 0.001).” mean in the caption of Figure 1?

5-      Seemingly, only one biomarker 8-isoprotane was involved throughout the test. Could the authors also provide data on another biomarker, or pharmacological and physiological indicators?

Author Response

Please refer to the following comments for formatting issues:

Comment 1: It was confusing to use both upper-case and lower case characters in the Title. Please correct.

Response 1: The title has been changed to uppercase characters.

Comment 2: The codes of subtitles were incorrect.

Response 2: The arrangement of subtitles has been corrected.

Comment 3: The fronts used in figures were not unified.

Response 3: Fonts used in the figures' titles` and their legends have been corrected.

Comment 4: The size of figures varied a lot.

Response 4: The sizes of figures in the manuscript have been adjusted accordingly.

Please refer to the following scientific comments:

Comment 1: “Percent match: 41%” was revealed by the iThenticate similarity system. Please keep in mind that ethical issues must be avoided. Maybe a total similarity index < 20% should be reached?

Response 1: We ran the Grammarly Plagiarism check on the revised manuscript and it showed a 22% similarity with all references in the manuscript included.

Comment 2: Please test the cytotoxicity of the reagents in retina cells.

Response 2: The reagent employed in the present study with a potential for significant cytotoxicity is hydrogen peroxide (H2O2).  We and others have used this agent for inducing oxidative stress in ocular tissues.  The concentration of H2O2 employed in the present study (100 µM) is much lower than that reported by other investigators such as Cui et al. (2017; 200 – 800 µM), Hu et al. (2020; 300 – 400 µM) and Wang et al. (2023; 300 µM) in retinal cells. We have included a statement in the Discussion Section to discuss the effectiveness of the lower concentration of H2O2 in causing toxicity (page 13, lines 7-10).

Comment 3: The H2S production kinetics of the compounds should be evaluated.

Response 3: The kinetics of H2S release from slow- and fast-releasing compounds both in vitro and in vivo have been studied by several investigators (Li et al., 2008; reviewed by Powell et al., 2018).  Our use of these compounds in the present study was based on published data of the kinetic profile of the H2S-releasing compounds.  We have included a statement in the Discussion Section to reflect this point (pages 13-14, lines 20-25).

Comment 4: What did the “Concentration (p < 0.03) Time (p < 0.0268) Interaction (p < 0.001).” mean in the caption of Figure 1?

Response 4: The above-listed information was a typographical error and has been deleted.

Comment 5: Seemingly, only one biomarker 8-isoprotane was involved throughout the test. Could the authors also provide data on another biomarker, or pharmacological and physiological indicators?

Response 5: While evidence available support the use of several markers to validate oxidative stress in biological samples (Frijhoff et al., 2015; Czerska et al., 2015), we selected 8-isoprostanes because of its direct relevance to measurement of oxidative stress in retinal tissues (Nourooz and Pereira, 2000). Other investigators have also used 8-isoprostanes as markers of oxidative in the eye (Cervellati et al. 2014; Wong et al. 2015).  We have included a statement to reflect our justification for using 8-isoprostanes as a marker of oxidative stress in the Discussion Section (page 12, lines 7-11).

Reviewer 2 Report

Comments and Suggestions for Authors

the submission by Bush et al describes their characterisation of the H2S biosynthesis pathway in bovine retinal extract upon H2O2 – induced oxidative stress, as modified by H2S-releasing compounds and respective pathway inhibitors, using 8-isoprostane as the readouts of lipid peroxidation.  the concept of retinal protection by H2S stems back to the seminal work of Hideo Kimura.  since then the neuroprotective benefits have been demonstrated in cell cultures, isolated retina and rodent models.  It is helpful to the field for further understanding on the therapeutic mechanism of H2S in retinal degenerative diseases.  

the authors should cite and discuss a recent review:  10.1016/j.exer.2023.109568

Author Response

Comment 1: the authors should cite and discuss a recent review:  10.1016/j.exer.2023.109568

Response 1: Thank you for your recommendation. We have revised the Introduction Section to include recent papers that demonstrate the neuroprotective actions of H2S in mammalian tissues or organs including the eye. The article by Cornwell and Badiei, 2023 was found to be relevant to the study and has also been included in this Section (page 5-6, lines 15-20; 22-23; page 6, lines 24-27).

Reviewer 3 Report

Comments and Suggestions for Authors

The manuscript investigates the response of a very sensitive tissue, the retina, to oxidative stress and its possible protective factors under different conditions. Many diseases of the retina includeoxidative stress as a possible pathogenic factor: e.g. AMD, DM, ROP.

The abstract is not structured and therefore difficult to follow. The last sentence of the conclusion isnot supported by anything, it is a hypothetical statement. 3 MST pathway is a special jargon, nobackground in the abstract.

Material, Methods, Tissue preparation

The retina is an extremely oxygen-sensitive tissue. (Hayreh's old experiments modelling arterial occlusion in Rhesus monkeys showed that irreversible damage to the retina develops after only 90 min. In humans, CRAO can only be reversed within 4 hours with intravenous rt-PA.)

Has histopathology been done e.g. in a fellow eye?

Please complete paragraph 1.1: How many pieces of retinal tissue were examined? Were both eyes of the same animal examined? What was the control eye? Was the entire retinal tissue used or just a piece?

Please add to chemical analyses: were the retinal tissues tested separately or in a group.

In the Discussion, the oxygen sensitivity of the retina should be analysed. Discuss why it is difficult to study retinal tissue in vitro. Explain why a time window of 24 hours is considered appropriate and not shorter e.g. 1-4hour?
Explain and demonstrate that bovine retinal tissue has not been subjected to
oxygen damage in the time between animal sacrifice and tissue harvesting.

Author Response

Comment 1: The abstract is not structured and therefore difficult to follow. The last sentence of the conclusion is not supported by anything, it is a hypothetical statement. 3 MST pathway is a special jargon, no background in the abstract.

Response 1: In response to the Reviewer’s Comments, we have revised the content of the Abstract for brevity.

Comment 2: The retina is an extremely oxygen-sensitive tissue. (Hayreh's old experiments modelling arterial occlusion in Rhesus monkeys showed that irreversible damage to the retina develops after only 90 min. In humans, CRAO can only be reversed within 4 hours with intravenous rt-PA.)

Response 2: In a study reported by Murali et al.(2020), histopathology of ex-vivo human retinal explants in culture was performed to assess changes to retinal tissue postmortem. These investigators found that human retinal tissues remained resilient to cell death postmortem, with a substantial number of cells remaining viable for up to 5 days. While the thickness of retinal tissues was found to be reduced by 50% after 24 hours, the different retinal layers were intact for up to three days in culture. In our studies, the enucleated bovine eyes were transported to the laboratory in buffer solution and placed in an ice bath. After dissection, tissues were immediately placed in a warm oxygenated buffer solution and allowed to equilibrate for 30 minutes before the commencement of experiments. Consequently, based on our experimental design, we didn’t expect any significant oxygen damage to our tissues in the time between animal sacrifice and tissue harvesting.  We have included this information in the Methods Section of the manuscript (page 18, lines 10-13).

Comment 3: Has histopathology been done e.g. in a fellow eye?

Response 3: We used isolated retinal tissues from bovine eyeballs for our studies.  We did not perform histopathology of these tissues because our endpoint was the measurement of oxidative stress and not apoptosis.

Comment 4: Please complete paragraph 1.1: How many pieces of retinal tissue were examined? Were both eyes of the same animal examined? What was the control eye? Was the entire retinal tissue used or just a piece?

Response 4: After the isolation of the retinae, explants were cut into four pieces and randomly assigned into treatment groups. Yes, both eyes of a particular animal may have been used but they were not identified as such because of the random assignment of tissues from multiple eyeballs. The control group received neither H2O2 nor H2S-releasing compounds. We have included this information in the Methods Section for clarification purposes (page 18, lines 8-10).

Comment 5: Please add to chemical analyses: were the retinal tissues tested separately or in a group.

Response 5: Retinal tissues were tested separately in different experimental groups for statistical purposes.

Comment 6: In the Discussion, the oxygen sensitivity of the retina should be analysed. Discuss why it is difficult to study retinal tissue in vitro. Explain why a time window of 24 hours is considered appropriate and not shorter e.g. 1-4hour?
Explain and demonstrate that bovine retinal tissue has not been subjected to oxygen damage in the time between animal sacrifice and tissue harvesting.

Response 6: In a study reported by Murali et al.(2020), histopathology of ex-vivo human retinal explants in culture was performed to assess changes to retinal tissue postmortem. These investigators found that human retinal tissues remained resilient to cell death postmortem, with a substantial number of cells remaining viable for up to 5 days. While the thickness of retinal tissues was found to be reduced by 50% after 24 hours, the different retinal layers were intact for up to three days in culture. In our studies, the enucleated bovine eyes were transported to the laboratory in buffer solution and placed in an ice bath. After dissection, tissues were immediately placed in a warm oxygenated buffer solution and allowed to equilibrate for 30 minutes before the commencement of experiments. Consequently, based on our experimental design, we didn’t expect any significant oxygen damage to our tissues in the time between animal sacrifice and tissue harvesting.  We have included this information in the Methods Section of the manuscript (page 18, lines 10-13).

Round 2

Reviewer 1 Report

Comments and Suggestions for Authors

I have no further questions.

Author Response

Reviewer has no further questions.